materials science

decay fungi, fungi growth, attenuated total reflectance, Fourier transform infrared, hydrogen bond, bond energy

**Author for correspondence:**
Barun Shankar Gupta
e-mails: bsbarun@gmail.com,
barun.gupta@ntnu.no

# *In vitro* cell composition identification of wood decay fungi by Fourier transform infrared spectroscopy

Barun Shankar Gupta, Bjørn Petter Jelle and Tao Gao

Department of Civil and Environmental Engineering, Norwegian University of Science and Technology (NTNU), Høgskoleringen 7A, 7491 Trondheim, Norway

BSG, 0000-0001-8819-4820

Decay pathogens follow dissimilar metabolic mechanisms to cause irreversible damage to woody tissues. The objective of this study is to perform inter- and intra-species microbial cell structural comparison using attenuated total reflectance Fourier transform infrared spectroscopy. Representative fungi species, causing brown rot and white rot, namely, *Postia placenta* and *Trametes versicolor*, respectively, were cultured in laboratory conditions. *In vitro* spectral measurements were performed at periodic two week intervals of fungal growth. The study shows structural differences for both species of fungi. The prominent presence of protein amide, carbohydrate and carboxyl bands was of interest. Spectral deconvolution of the infrared broadband around approximately 3300 cm$^{-1}$ produced peaks at four different wavenumbers. The hydrogen bond energy obtained at the four wavenumbers, from deconvolution, varied from approximately 41 kJ mol$^{-1}$ to approximately 7 kJ mol$^{-1}$, indicating the presence of strong and weak forces in microbial cell structure. The hydrogen bond distance, obtained at the deconvoluted wavenumbers, varied between 2.7 Å–2.8 Å, indicating the presence of short and long-distance forces within microbial cells. Microscopic observation showed mycelium colonization, hyphal tip and lateral branching.

## 1. Introduction

Pathogens, like fungi, decompose forest wood thereby contributing to the global carbon cycle [1,2]. Chemically, wood degradation implies breaking of chemical bonds of the wood cellulose, hemicellulose and lignin components [3] at high moisture content [4]. In natural outdoor conditions, the fungal degradation of wood is a slow process and often does not show any visible discoloration, at the onset, causing difficulty in judgement of the soundness of wood. Contrarily, in accelerated laboratory

conditions, using—(i) sterile materials, (ii) nutrients, (iii) monoculture, and (iv) moisture, the interspecies antagonistic invasion is prevented thereby allowing the microbial cells to follow normal metabolic and enzyme activities. A study of fungal cell structure in accelerated laboratory conditions, therefore, provides meaningful information about the molecular compositions in microbial cell structure [5]. However, the available knowledge on spectroscopic study primarily deals with the chemical changes occurring in the degraded wood substrate, and not on the fungi cell structure [5–7].

Attenuated total reflectance (ATR) Fourier transform infrared (FTIR) spectroscopy is used in this study to collect molecular information of microbial cell compositions. ATR-FTIR is being widely used in forensic sciences for surface characterization of biomaterials, polymers and microbes [5,8–10]. In this study, the white rot fungus *Trametes versicolor* (Fr.) Phil. (Basidiomycota) and brown rot fungus *Postia placenta* (Fr.) M.J. Larsen & Lombard 1986 (Basidiomycota) were cultured in laboratory conditions. Both species of fungi were cultivated on Norway spruce wood (*Picea abies* [L.] Karst.). Norway spruce is a widely available tree in northern Europe.

The periodic sampling design of harvesting fungi in two week intervals, was adopted from previous research work, in which decayed samples of beech wood (*Fagus sylvatica*) were exposed to the white rot fungus *T. versicolor* and was analysed by using ATR-FTIR spectroscopy [11]. The authors [11] compared the infrared bands obtained from the wood substrate at different stages of fungal attack with the non-decayed wood and observed appearance of new bands after four weeks of exposure, which was a prominent indication of new bands obtained from altered cell wall constituents owing to reaction with fungal enzymes. In addition, subtraction of bands obtained from wood decayed after 70 and 84 days showed that *T. versicolor* has the ability to remove all types of wood cell-wall constituents, i.e. lignin, hemicelluloses and cellulose [11]. Enzyme properties are areas of recent research owing to their communication abilities [12]. However, it was observed that holocellulose (cellulose and hemicellulose), rather than lignin, are preferentially degraded by *T. versicolor* [13].

Molecularly, the wood fungal cell wall is composed of structural and intra-structural components [14]. The biosynthesis of the cell wall components of fungi is a complicated biological process. Glucan and chitin are synthesized at the plasma membranes that get extruded into the cell wall giving structural integrity to the fungal cell. Glucan is the major structural polysaccharide comprising around 50–60% of the cell wall by dry mass [14–17]. Chitins are beta-1, 4-linked *N*-acetylglucosamines, which account for 2–20% of the dry mass of the cell wall components. In addition, there are glycoproteins that constitute 20–30% of cell wall components controlling the dynamics of the cell wall structure by breaking and forming bonds between the cell wall polysaccharides [15]. Bond energies provide valuable information to solve the intricacies of molecular structure [18]. Structural components and proteins, in a cell, are extensively hydrogen bonded. Intramolecular hydrogen bond energies in the range of 1.4–13.7 kcal mol$^{-1}$ provide strength for aliphatic structures [19].

The fungi, *T. versicolor* and *P. placenta*, follow dissimilar mechanisms to degrade wood. It has been observed that the fungus, *P. placenta*, secretes extracellular reactive oxygen species for initial depolymerization of cellulose in wood [20–24]. The protein content in the enzyme supernatant from the substrates for both *T. versicolor* and *P. placenta* remains fairly constant over 16 weeks of evaluation [25]. Contrarily, the amount of glucose decreases for *T. versicolor*. However, glucose has a fluctuating up and down effect for *P. placenta* substrates. *Postia placenta* produces endo-1,4-glucanase activity that is 10 times higher than the activity by *T. versicolor* [25]. Henceforth, it can be said that several polymers perform active roles in fungal cell activities.

From the above discussions, a hypothesis can be drawn that the cellular structure of wood fungi depends on the duration of the wood deterioration process and 70–84 days of microbial exposure is sufficient for the wood cell wall material to get degraded. Henceforth, the aim of this study is to perform inter-species investigation of cell structure during the eight-week period of the wood decay process through monoculture. In order to investigate the cell structure rigidity, *in vitro*, the hydrogen bond energy and hydrogen bond distance from the cell structure materials and protein were measured.

# 2. Experimental

## 2.1. Materials

Two microbiological culture media were selected to help fungal growth on agar plates. The white rot fungus, *T. versicolor* (strain CTB 863A), was cultivated on agar plates containing 20 g Fluka malt, 10 g Fluka agar (4%) and 500 ml water. The brown rot fungus, *P. placenta* (strain FPRL 280), was cultivated

on agar plates containing 20 g Bacto malt, 10 g NMD agar (4%, NMD = Norsk Medisinaldepot AS) and 500 ml water at the Norwegian Forest and Landscape Institute, Norway. A total of 64 specimens, which were cubic blocks, of Norway spruce wood (*Picea abies* [L.] Karst.), of dimension: 5 mm × 10 mm × 30 mm, were dried at 103°C for 18 h and dry masses were recorded for each block. The oven-dry specimens were conditioned in room temperature and were set aside for sterilization. Sterilization of the acclimatized specimens was performed at 121°C for 20 min. Statistical analysis was performed to check the variation in mass of wood blocks.

## 2.2. Fungi culture method

Pairs of sterilized wood blocks were placed on each agar plate for each type of fungi as substrates. Half of the agar plates (16 nos.) were inoculated with the strains of brown rot fungus, *P. placenta*, and the other half (16 nos.) were inoculated with the strains of white rot fungus, *T. versicolor*. In total, four plates, each containing two wood blocks (total eight wood blocks) inoculated by fungi were evaluated after 2, 4, 6 and 8 weeks of inoculation as followed by previous researchers [11,24,26]. The plates were sealed by adhesive tape to prevent contamination. During the culture duration, fungi hyphae grew out of the agar on the wood substrates. Mycelium of each fungal strain was collected from the thick mycelial film grown on wood blocks by using a pair of sterilized tweezers, in a bio-hood. After collection, the remaining mycelium film from all wooden blocks was wiped clean by gently rubbing with soft tissue papers without removing wood fibres at each evaluation period. Masses of all samples were recorded immediately after the removal of mycelium. One wood specimen from each agar plate was dried for 18 h at 103°C and stored in an air-tight desiccator at ambient conditions. As this study focuses on the cellular activities of fungi, the mass loss that occurred in the wood blocks, during observation periods is not provided in this article.

## 2.3. Attenuated total reflectance-Fourier transform infrared spectroscopy

Though spectral information was collected *in vitro*, the moisture conditions of the fungal cell were not altered, i.e. living cell characterization was performed simulating normal growth atmosphere. The study assumes that the collection of spectral information *in vitro* did not alter the microbial cell material compositions within the characterization time frame.

The FTIR material characterization at different periods of fungal growth was performed with a Thermo Nicolet 8700 FTIR spectrometer having a Smart Orbit accessory, i.e. a horizontal ATR accessory (single reflection) with a diamond crystal, wavelength range 4000 cm$^{-1}$ (2.5 µm) to 400 cm$^{-1}$ (25 µm), in an atmosphere with minimalized $CO_2$ and $H_2O$ content through purging by a Parker Balston 74-5041 FTIR purge gas generator. With the help of a pair of sterilized tweezers, a thin film of mycelium was placed on the top of the diamond crystal of the ATR-FTIR accessory at ambient conditions. Mechanical pressure from a rotating knob was applied on the top of the specimen to acquire adequate contact with the diamond crystal, i.e. without air pockets at ambient conditions of room temperature and room humidity. Each FTIR spectrum presented is based on a recording of average of 32 scans at a resolution of 4 cm$^{-1}$. The FTIR spectra presented in this study have not been ATR corrected, neither with respect to penetration depths nor absorbance band shifts, which are dependent on the refractive indices of the sample and the ATR crystal (diamond in this case) and the angle of incident radiation. Because the focus of the study was to compare the information from original ATR-FTIR spectra, no ATR corrections were performed. Besides, the raw ATR-FTIR data, in either transmittance or logarithmic absorbance mode, is usually preferred to perform comparative analysis. It may be noted that errors may get introduced in the ATR corrected spectra for erroneous input of assumed refractive index data. Four replicates of each fungus at each evaluation period were recorded. OMNIC software was used for spectral analysis including spectral derivative and deconvolution. MINITAB 16.2.2 was used for statistics.

The hydrogen bond energies from H-N and H-O groups [27,28] were calculated by using the following equation (2.1):

$$E_H = \frac{1}{K} \frac{\nu_0 - \nu}{\nu_0},$$

(2.1)

where $\nu_0$ = standard frequency of the free –OH group at 3650 cm$^{-1}$, $\nu$ = frequency of the bonded –OH group and K = a constant equal to $(1/2.625) \times 10^{-2}$ (kJ mol$^{-1}$)$^{-1}$.

**Table 1.** Analysis of variance (2-way ANOVA) of dry mass of wood block substrates, used in fungi culture.

| si. no. | source of variation | SS | d.f. | MS | F | p-value | F crit |
|---|---|---|---|---|---|---|---|
| 1 | sampling of 4 harvesting patterns of fungi at 2nd, 4th, 6th and 8th weeks of growth | 0.0048 | 3 | 0.0016 | 0.6778 | 0.5693 | 2.7694 |
| 2 | species (two fungi, *Postia* sp. and *Tametes* sp.) | 0.0053 | 1 | 0.0053 | 2.2234 | 0.1415 | 4.0130 |
| 3 | interaction between rows 1, 2 | 0.0024 | 3 | 0.0008 | 0.3392 | 0.7971 | 2.7694 |
| 4 | interaction within rows 1, 2 | 0.1324 | 56 | 0.0024 | | | |
| 5 | total | 0.1449 | 63 | | | | |

To further explore the hydrogen bonds centred on the 3300 cm$^{-1}$ regions, the hydrogen bond distances were calculated [29] from the Sederholm equation:

$$\Delta v = 4.43 \times 10^3 \, (2.84 \, - \, R), \tag{2.2}$$

where $\Delta v = v_0 - v$, $v_0$ = monomeric OH stretching frequency at 3600 cm$^{-1}$, $v$ = stretching frequency observed in the infrared (IR) region and R = hydrogen bond distance in Å (1 Å = $10^{-10}$ m). OMNIC software was used for spectral analysis. The bands of the nearest frequencies were marked in the enlarged spectra. Between each spectrum, there were differences of 50–100 cm$^{-1}$. The averages of the energy values and bond distances were thus obtained.

## 2.4. Scanning electron microscopy

In order to characterize a thin layer of mycelium film a piece of one-sided adhesive tape was lightly touched on the top of fungal culture/colony to collect hyphae on adhesive tape [30]. Scanning electron microscopy (SEM) images, from the S-3400 N Hitachi make, was collected at an accelerating voltage of 15 kV. Back scatter images were captured, to avoid charge build-up on the film, for better representation.

# 3. Results and discussion

The wooden blocks of Norway spruce were used as nutrient source to fungi. Wood inoculated with the strains of *T. versicolor* and *P. placenta*, were weighed at two week intervals (table 1). Before sterilization, the wood specimen masses were in the range of 0.8–0.9 g. After oven drying, the dry masses of the wood specimen were in the range of 0.8 g. The average green mass and the dry mass of the control wood blocks were 0.8 and 0.7 g, respectively. Table 1 shows the analysis of variance (ANOVA) of the dry mass of wood blocks. From table 1, it can be statistically inferred with 95% confidence that there is no significant variance (*p*-value > 0.05) in dry mass of wood blocks that were used as substrates, or source of nutrients, to fungi. Therefore, it can be assumed that for the experimental growth conditions and fungi species, similar nutrients were used as substrates. Statistical paired *t*-test was performed by grouping 32 data of wood mass, for eight replicates at four growth intervals, for each set of fungi. The *t*-test output (*p*-value > 0.05, two-tail), at the 95% confidence limit indicates that there is no statistically significant relationship in the rate of decay loss of wood substrate caused by two different species of fungi.

## 3.1. Fourier transform infrared spectra

The average FTIR spectra obtained by scanning the cellular structure of the *T. versicolor* and *P. placenta* fungi specimens are represented in figures 1–4. The IR spectra displayed several regions of interest, namely, the region around 3330 cm$^{-1}$ representing the H-bonded hydroxyl (OH) and NH groups, the region around 1636 cm$^{-1}$ representing the amide bands, and the region around 1030 cm$^{-1}$ representing the primary-OH from cellulose (CO) [31,32]. The NH stretching vibrations of amide A and amide B at approximately 3300 cm$^{-1}$ and approximately 3170 cm$^{-1}$, respectively, are prominent owing to the polypeptide backbone and hydrogen bonding [31,32]. Hydrogen bonding may also occur between the water molecule and –OH group of side chains [33]. Note that water has a broad band above 3000 cm$^{-1}$ and a sharper band around 1640 cm$^{-1}$, which possesses identification challenge owing to the overlapping bands with various organic IR bands [34]. However, owing to the presence of

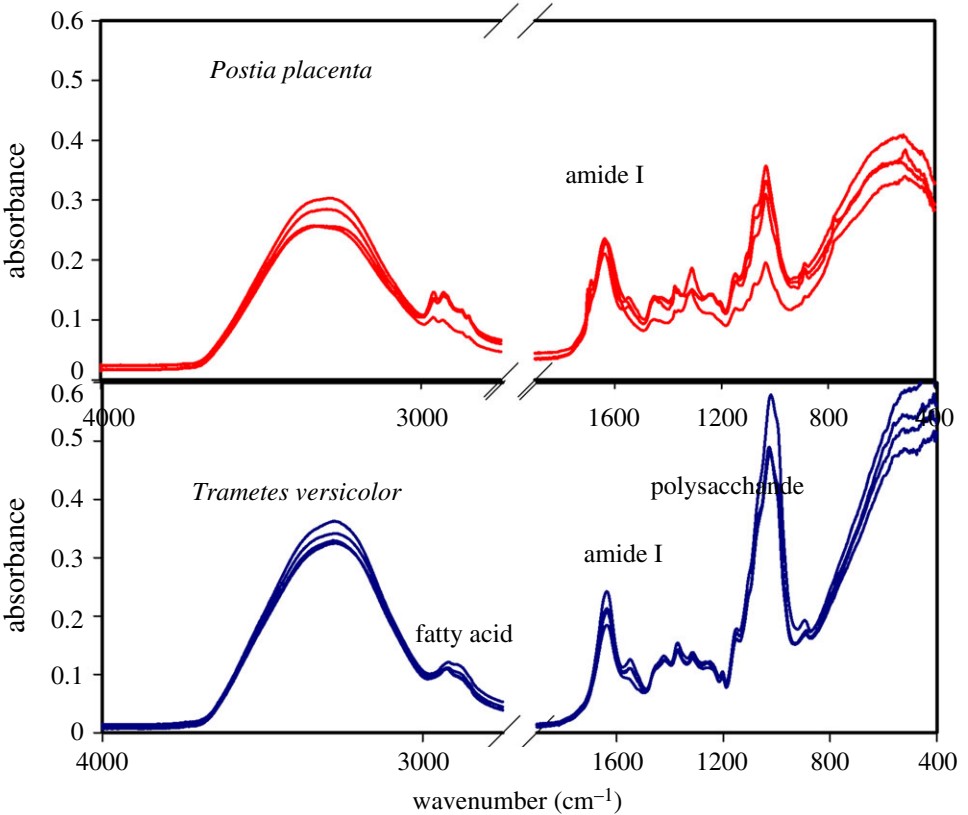

**Figure 1.** Absorption spectra of *Trametes versicolor* and *Postia placenta* for 6th week of growth, obtained from fungi grown on spruce wood (*Picea abies*) substrate, showing spectral reproducibility in the regions of (*a*) fatty acid, (*b*) amide I and (*c*) polysaccharides.

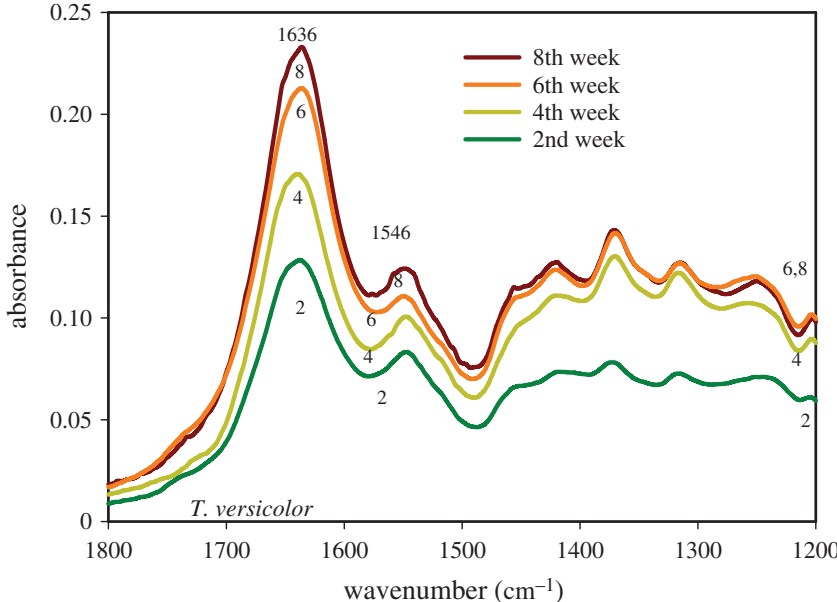

**Figure 2.** FTIR spectra of the 'fingerprint region' of *Trametes versicolor* after different periods of growth on spruce wood (*Picea abies*) substrate.

a double-triple bond character in CO in protein, the absorption of CO in the IR region is much sharper than the O-H bond [35,36]. Consequently, the stretching frequency of the C-O bond will be sharper and stronger than the adsorbed moisture. ATR-FTIR has fixed beam spot size. The measurements were performed in room conditions and room humidity. Considering the fact that four replicates were scanned, it can be assumed that the small-sized specimens had time enough to equilibrate into ambient room conditions.

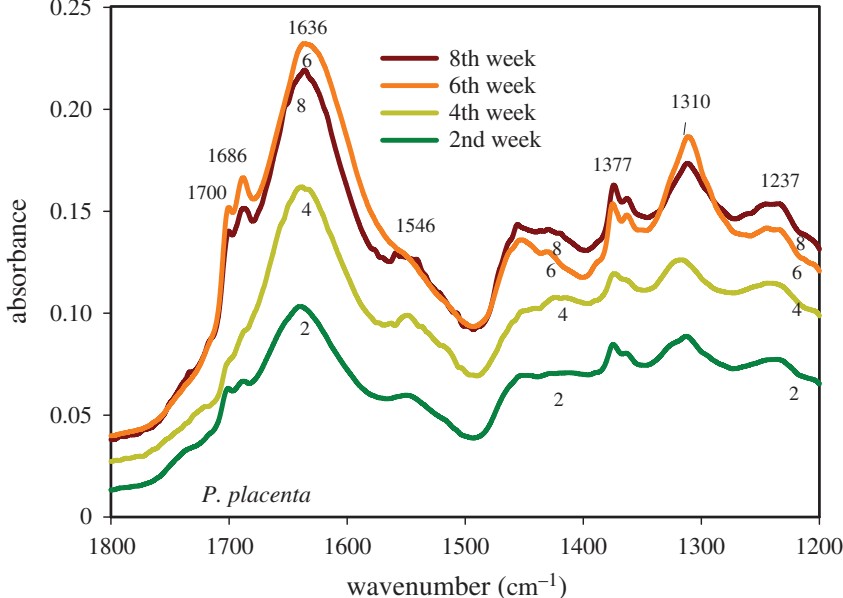

**Figure 3.** FTIR spectra of the 'fingerprint region' of *Postia placenta* after different periods of growth on spruce wood (*Picea abies*) substrate.

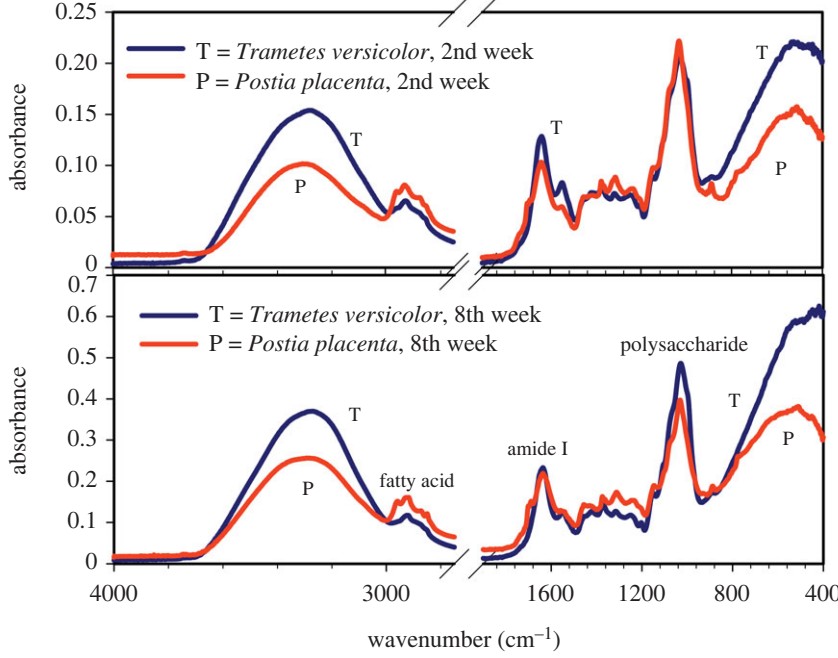

**Figure 4.** FTIR spectra of *Trametes versicolor* and *Postia placenta* after two weeks and eight weeks of growth on spruce wood (*Picea abies*) substrate.

In this work, the IR band at 1636 cm$^{-1}$ is assigned to amide I for C=O stretching vibrations with contributions from the out-of-phase CN stretching vibration, the CCN deformation and the NH bending of protein [36–39]. The band at 1546 cm$^{-1}$ is assigned to amide II for N-H in-plane bending and the CN stretching vibration of protein [37–39]. The amide I and amide II bands are hardly affected by the side chain and is commonly used for secondary structure analysis [38]. The band at 1400–1200 cm$^{-1}$ for CN stretching and N-H bending with contribution from CO in-plane bending is assigned to amide III [38]. Cell lipids show a band at 1465 cm$^{-1}$ for CH$_2$ bending mode, a band at 1402 cm$^{-1}$ for C=O symmetric stretching of COO$^-$ groups and the band at 1377 cm$^{-1}$ for C-H bending mode of CH$_2$ [32,40]. The skeletal stretching vibrations are assigned to the region of 1200 cm$^{-1}$– 880 cm$^{-1}$. The band at 1030 cm$^{-1}$ is assigned to the C-O deformation in primary alcohols as complex bands resulting from C-O and C-O-C stretching vibrations [32]. Phospholipids and PO$_2$ show bands at

1237 cm$^{-1}$ and 1082 cm$^{-1}$ for asymmetric stretching and symmetric stretching vibrations, respectively [32,40]. These bands were observed in the structures of *T. versicolor* and *P. placenta*.

Figure 1 shows absorbance spectra obtained from the cell structure of *T. versicolor*. The broad absorbance for the hydrogen-bonded OH group at 3300 cm$^{-1}$ indicates that the structural components within the fungal cell show high affinity towards hydrogen bonding, especially with water [33]. The influence of water on bio-surfaces is interesting. The FTIR spectra from the cell structure of *P. placenta* show a similar tendency, i.e. there is a widening of peak in band absorbance centring around 3300 cm$^{-1}$. There is strong absorbance in the amide regions and at wavenumber 1030 cm$^{-1}$.

Figure 2 shows the higher resolution image of spectra of the fungal cell material within the range 1800–1200 cm$^{-1}$ for *T. versicolor*. A steady increase in absorbance of the amide I band at 1636 cm$^{-1}$ indicates increase in protein content in the fungal cell material from the 2nd week of growth to the 8th week of growth. This may be owing to the reason, that as the fungal colonization increases on the wood substrates, the invading microbes get more nutrient access compared to the starting period because of the process of enzymatic degradation. A strong band, $\nu$(C=C), is observed for the side chain of histidine amino acid, 4-methylimidazole, at 1633 cm$^{-1}$ [38]. Additionally, the protein amino acid, arginine, shows a band near 1635 cm$^{-1}$ for symmetric stretching vibration, $\nu_s(CN_3H_5^+)$, depending on the salt bridge between arginine and other residues [38]. There are differences in intensities of absorbance band for the amide II band at 1546 cm$^{-1}$, indicating an increase in protein content with growth period. Other researchers have also observed increase in protein content in fungal cell from FTIR studies and concluded that protein increase is related with cell maturity [37].

The expanded IR region in figure 3, for spectra obtained from *P. placenta*, shows increasing absorbance of the amide I band with increasing period of fungal growth. The absorbance of the band at 1310 cm$^{-1}$ for amide III, after the 6th week of growth is high compared to the growth patterns of the other three curves. Similarly, the absorbance of the band at 1636 cm$^{-1}$ for amide I, C=O stretching, for the 4th week of growth is high compared to the growth patterns of the 2nd week growth curve. There are two shoulders at 1700 cm$^{-1}$ and 1686 cm$^{-1}$, respectively, which are assigned to the carboxyl group from acids since *P. placenta* is a known species for secreting an acid sheath to surroundings. The band could also result from the protein amino acid, aspartic acid (C=O), indicating hydrogen bonding to the hydroxyl group [38]. There is a weak absorbance at 1546 cm$^{-1}$ and loss of protein amide II band centring at 1546 cm$^{-1}$ in the 6th week, which is contrary with the observation of figure 2, of *T. versicolor*. This indicates that there could be lesser influence of the protein amide N-H molecules. Additionally, this may indicate that the spectra were collected from the hyphal tip where the concentration of protein is less, as was found by other researchers [37]. This also shows the sensitivity of ATR-FTIR towards fungal cells. At lower wavenumber, there is an increase in absorbance of the C-H band of carbohydrates.

The comparative spectra for the growth of the two types of fungi are shown in figure 4. The broadness of the band at 3300 cm$^{-1}$ indicates the presence of higher moisture in the cell structure of *T. versicolor* compared to the cell structure of *P. placenta* after the 2nd week and 8th week of observations during fungal growth. The spectra from both species of fungi show sharpness in IR bands for fatty acid, amide I and polysaccharide regions.

## 3.2. Second derivative spectra

In order to further investigate the spectral features, second derivatives of the spectral region 1800–900 cm$^{-1}$ were performed and plotted graphically as shown in figures 5 and 6. Assigning bands to secondary protein structure is difficult [34]. Previous research has shown that there are two solvent media that were successfully used for spectral evaluation of protein, $H_2O$ and $D_2O$ media [34]. However, it has been observed that proteins in $H_2O$ provide much more reliable data than in $D_2O$ media, as the latter changes the protein property owing to exchange of H-D in peptide linkages [34]. In addition, the strength and length of the hydrogen bond might get altered by H-D exchanges in protein secondary structure for $D_2O$ media [34]. Therefore, we believe that the spectral information obtained in this study in natural conditions, i.e. $H_2O$, are representations of true O-H bond structure.

Previous researchers have assigned IR bands to the second derivative of receptor protein FTIR spectra as: α- helices at 1653 cm$^{-1}$, β-sheets near 1637 cm$^{-1}$, random coil at 1648 cm$^{-1}$ [34]. Table 2 shows the assignment of bonds to IR bands.

### 3.2.1. Molecular structure after two weeks of growth

Figure 5 shows the second derivatives of FTIR spectra of both types of fungi after two weeks of growth. The spectral resolution is higher in derivative spectra providing a clearer insight into IR bands. The

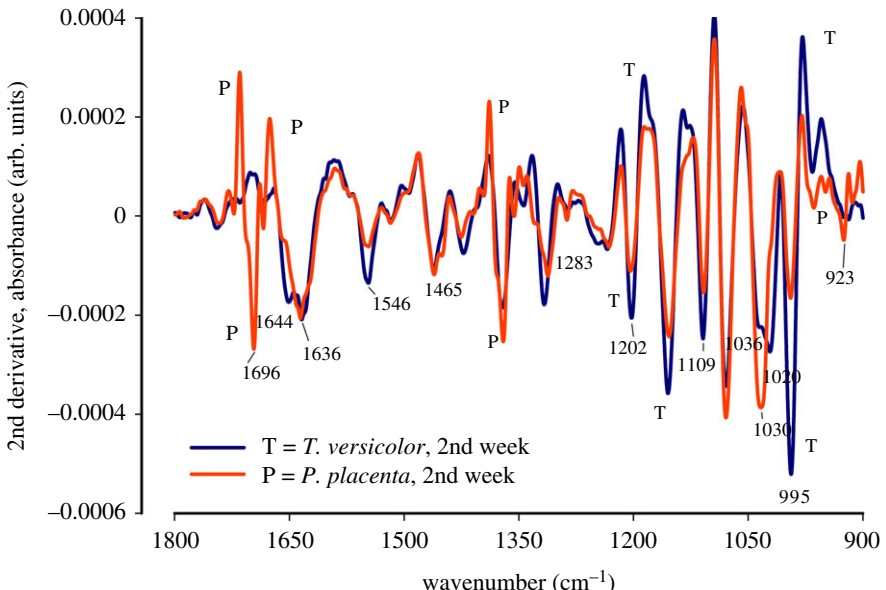

**Figure 5.** The second derivative FTIR spectra of *Trametes versicolor* and *Postia placenta* after two weeks of growth on spruce wood (*Picea abies*) substrate.

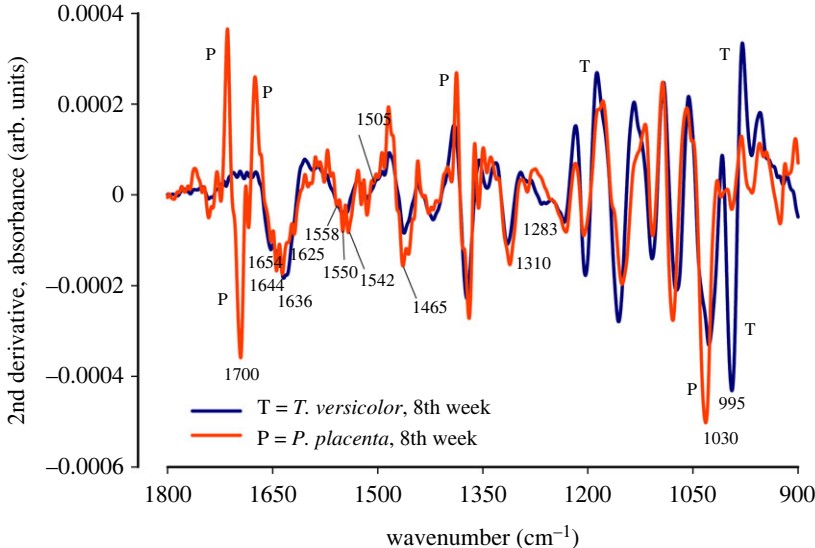

**Figure 6.** The second derivative FTIR spectra of *Trametes versicolor* and *Postia placenta* after eight weeks of growth on spruce wood (*Picea abies*) substrate.

amide I, the most important band in this study, is prominent in figure 5. The spectral features from *T. versicolor* were prominent. Absorbance bands in the original spectra are shown as negative bands in the second derivative spectra. The splitting of the amide I spectrum shows appearance of a band at 1636 cm$^{-1}$ and 1644 cm$^{-1}$ that can be assigned to the ß-sheet and random coils, respectively, of *T. versicolor*. The protein amino acids absorbance bands for, histidine, $v$(C=C), at 1633 cm$^{-1}$ and salt-bridged arginine, $v_s$(CN$_3$H$_5^+$), at 1635 cm$^{-1}$, co-appear in the same region [38]. The α-helices provide a primary absorption band at 1655 cm$^{-1}$ that shifts to lower wavenumber with increasing helix length [38]. The ß-sheet is found as a strong band near 1630 cm$^{-1}$ that shifts to higher wavenumber with twisting and reduces the splitting between two bands [38]. At lower wavenumber, the bands at 1283 cm$^{-1}$ is assigned to α-helices of the amide III absorption band owing to in-phase combination of CH stretching and NH bending vibrations [42]. The second derivative spectra show band splitting of the carbohydrate 1030 cm$^{-1}$ band of *T. versicolor* into two: (i) one at 1036 cm$^{-1}$ showing presence of carbohydrate C-C bond and (ii) the other at 1020 cm$^{-1}$ indicating primary alcoholic $v$(C-O), both are probably from accumulated cellulose owing to thickening in the cell wall.

**Table 2.** IR band assignment from second derivative spectra obtained from figures 5 and 6.

| species | growth (weeks) | structure | band position | assignments |
|---|---|---|---|---|
| *Trametes versicolor* | 2nd | amide | 1644 cm$^{-1}$ | hydrogen bonded long α-helix [41] |
| | | | | ß-sheet [34] |
| | | | 1636 cm$^{-1}$ | twisted ß-sheet [34,41] |
| | | | | hist, $\nu$(C=C) [33] |
| | | | | salt bridged Arg, |
| | | | | $\nu_s$(CN$_3$H$_5^+$) [34] |
| | | carbohydrate | 1036 cm$^{-1}$ | carbohydrate C-C bond [41] |
| | | | | poly-β-L-Ala |
| | | | 1020 cm$^{-1}$ | primary alcohol, $\nu$(C−O) [41] |
| *Postia placenta* | 8th | carboxyl | 1700 cm$^{-1}$ | keto C=O [33] |
| | | amide | 1654 cm$^{-1}$ | α-helix [34,41] |
| | | | | Arg, $\nu_{as}$(CN$_3$H$_5^+$) |
| | | | 1644 cm$^{-1}$ | disordered amide I band [41] |
| | | | | Arg, $\nu_s$(CN$_3$H$_5^+$) |
| | | | | ß-sheet [34] |
| | | | 1625 cm$^{-1}$ | ß sheet [34,41] |
| | | | | Lys, $\delta_{as}$(NH$_3^+$) [34] |
| | | | | poly-β-L-Glu |
| | | | 1558 cm$^{-1}$ | Glu, $\nu_{as}$(COO$^-$) [41] |
| | | | 1550 cm$^{-1}$ | Asp, $\nu_{as}$(COO$^-$) [41] |
| | | | | amide II, NH bending |
| | | | 1542 cm$^{-1}$ | Asp, $\nu_{as}$(COO$^-$) [41] |
| | | | | Glu, $\nu_{as}$(COO$^-$) |
| | | | 1283 cm$^{-1}$ | amide III, NH bending [42] |
| | | | 1310 cm$^{-1}$ | |
| | | carbohydrate | 1030 cm$^{-1}$ | primary alcohol, $\nu$(C−O) [41] |
| | | | 1505 cm$^{-1}$ | lignin [6] |
| | | | | Trp, $\nu$(CN), $\delta$(CH), $\delta$(NH) [41] |

### 3.2.2. Molecular structure after eight weeks of growth

The second derivative of the amide I (1636 cm$^{-1}$) and amide II bands are shown in figure 6. The IR band from *P. placenta*, in figure 6, after eight weeks of growth, shows splitting into three bands, suggesting occurrence of α-helices at 1654 cm$^{-1}$, disordered amide I band at 1644 cm$^{-1}$ and ß sheet at 1625 cm$^{-1}$ [38]. The bending vibration of protein bound water at 1640 cm$^{-1}$ is often strong [38]. The strength and sharpness of the IR band at this region, therefore, is a good indicator of presence of protein groups. The absorbance of the band at 1644 cm$^{-1}$, after eight weeks of growth, is relatively stronger than the 1644 cm$^{-1}$ band absorbance after two weeks of growth. This suggests that the structure of protein might have changed with an increasing period of growth. The antisymmetric band, $\nu_{as}$(CN$_3$H$_5^+$), for amino acid arginine shows presence in the band region of 1695–1652 cm$^{-1}$ [38]. The symmetric band, $\nu_s$(CN$_3$H$_5^+$), of amino acid arginine shows presence in the band region of 1663–1614 cm$^{-1}$ [38]. The asymmetric band, $\delta_{as}$(NH$_3^+$), of protein amino acid lysine, shows presence in the band region of 1629–1626 cm$^{-1}$ [38]. At lower wavenumber, the bands at 1310 cm$^{-1}$ and 1283 cm$^{-1}$ are assigned to α-helices and ß-sheet of amide III band owing to in-phase combination of CH stretching and NH bending vibrations [42]. However, contrary to figure 5, there is no split in the band at 1030 cm$^{-1}$, suggesting dominant presence of the primary alcohol C–OH group, which possibly originated from

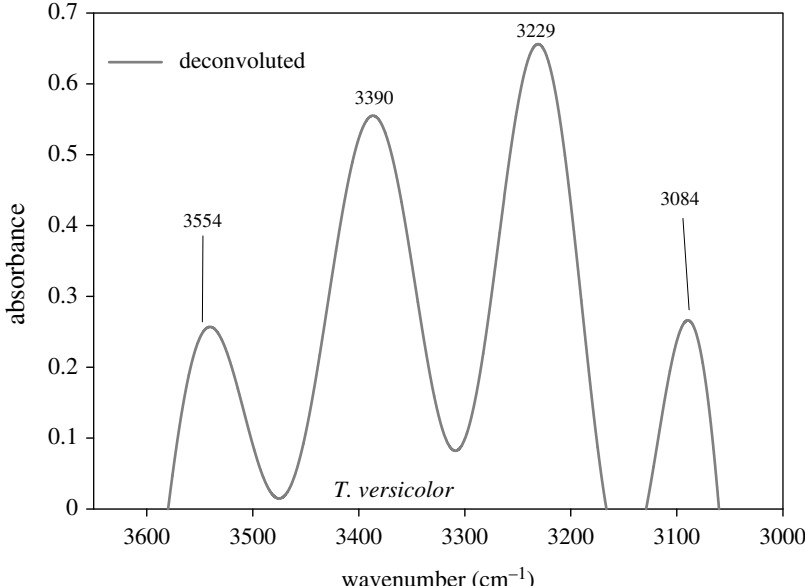

**Figure 7.** The deconvolution of FTIR spectra of *Trametes versicolor* after two weeks of growth on spruce wood (*Picea abies*) substrate.

the cellulosic C–OH group. There is a small and sharp band at 1505 cm$^{-1}$ in the second derivative spectra of *P. placenta*, indicating the possible presence of lignin; or, the band at 1505 cm$^{-1}$ could be from the bands of stretching vibrations, $v$(CN), $\delta$(CH), $\delta$(NH), of the amino acid tryptophan, that is found at 1509 cm$^{-1}$ [38]. Besides, the amide II band (1546 cm$^{-1}$) from the spectra of the *P. placenta* cell is split into three bands (1558 cm$^{-1}$, 1550 cm$^{-1}$, 1542 cm$^{-1}$) suggesting the possible presence of the α-helices and ß-sheet structure of cell proteins. The antisymmetric band, $v_{as}$(COO$^-$), of the protein glutamic acid, is observed in the region 1560–1556 cm$^{-1}$ [38]. Furthermore, the asymmetric vibration, $v_{as}$(COO$^-$), of the protein aspartic acid, is observed in this band region upon cation chelation [38].

Comparing figures 5 and 6, it can be said that the spectral information in figure 6 (after eight weeks of growth) is from matured *P. placenta* cells having high concentration of protein and cellulose in their cell structure. The increase in protein content and carbohydrate content may also signify the spore formation or germination [37] after eight weeks of growth.

### 3.2.3. Fourier deconvolution

Fourier deconvolution is an easy process to discover the bands that overlap in case of a broader absorption band in the IR spectra [33]. The IR band region of 3760–3630 cm$^{-1}$ is assigned to the symmetric and antisymmetric stretching of non-hydrogen bonded water [38]. Contrarily, the IR region 3600–2900 cm$^{-1}$ provides information about a bonded OH group of water depending on the strength of hydrogen bonding [38]. Furthermore, the IR spectral region of 3700–3100 cm$^{-1}$ closely relates with the O-H stretching vibration of water molecules [43]. It is assumed that in the water imbibed cell structure, 50% of the N-H and C=O groups are hydrogen bonded with water [38]. Depending on the strength of the hydrogen bond, the NH stretching vibration from amides produces bands at 3310 cm$^{-1}$ and 3270 cm$^{-1}$ that are localized on the NH group [38]. The band at 3297 cm$^{-1}$ is assigned to amide A of protein [44].

The 3600–3050 cm$^{-1}$ region, which shows a broad band in the FTIR spectra for both species of studied fungi, on deconvolution, represents the presence of four bands at 3554 cm$^{-1}$, 3390 cm$^{-1}$, 3229 cm$^{-1}$ and 3084 cm$^{-1}$, as shown in figure 7. The deconvolution spectra show similarity in information obtained from two species of fungi, for the different growth period. The four bands (OH), obtained in figure 7, from high to low wavenumbers, can be assigned to OH stretching vibrations of: (i) free water ($v_{as}$ stretching), (ii) free water ($v_s$ stretching), (iii) cluster water not fully coordinated ($v_{as}$), and (iv) tetrahedral coordinated water ($v_s$ stretching) [43]. The intensity of the IR band of $v_{as}$ stretching of free water is weak, which leads to the conclusion that most of the water molecules, accessed by ATR crystal, within the microbial cell is hydrogen bonded.

**Table 3.** Hydrogen bond energy ($E_H$, kJ mol$^{-1}$) at different wavenumbers (cm$^{-1}$) at shifting frequencies at various growth times.

| species | growth (weeks) | hydrogen bond energy (kJ mol$^{-1}$)[a] | | | |
|---|---|---|---|---|---|
| | | 3554 cm$^{-1}$ | 3390 cm$^{-1}$ | 3229 cm$^{-1}$ | 3084 cm$^{-1}$ |
| *Trametes versicolor* | 2 | 7.2 ± 0.1 | 18.8 ± 0.3 | 30.0 ± 0.4 | 40.5 ± 0.2 |
| | 4 | 6.9 ± 0.3 | 18.5 ± 0.2 | 30.3 ± 0.3 | 40.5 ± 0.2 |
| | 6 | 6.8 ± 0.7 | 18.4 ± 0.2 | 29.8 ± 0.9 | 40.4 ± 0.2 |
| | 8 | 6.6 ± 0.4 | 18.5 ± 0.1 | 30.4 ± 0.2 | 40.6 ± 0.3 |
| *Postia placenta* | 2 | 7.2 ± 0.4 | 19.3 ± 0.3 | 30.4 ± 0.6 | 40.5 ± 0.3 |
| | 4 | 7.2 ± 0.1 | 19.0 ± 0.3 | 29.9 ± 0.7 | 40.2 ± 0.1 |
| | 6 | 6.8 ± 0.3 | 18.5 ± 0.3 | 30.2 ± 0.1 | 40.2 ± 0.3 |
| | 8 | 7.2 ± 0.1 | 18.8 ± 0.5 | 29.8 ± 0.4 | 40.1 ± 0.1 |

[a]Mean of four measurements ± 2 s.d. of the mean (95.45% confidence interval).

## 3.4. Hydrogen bond energy

Deconvolution and IR peak finding allowed calculation of the bond energy of the hydrogen bonds (equation (2.1)) that were present in the broadband 3600–3050 cm$^{-1}$ region. The hydrogen bond (H-bond) energies for fungal growth at the 2nd, 4th, 6th and 8th weeks are given in table 3. The H-bond energies, from discussions in §3.3., can be associated with the presence of water molecules in the cell structure. Consequently, table 3 provides an idea that the H-bond at 3554 cm$^{-1}$ and 3390 cm$^{-1}$, which might have been influenced by the weak signal from free water OH ($v_s$ and $v_{as}$ stretching) is low in magnitude (approx. 7.2 kJ mol$^{-1}$ and approx. 18.5 kJ mol$^{-1}$). Contrarily, the H-bond at 3229 cm$^{-1}$ and 3084 cm$^{-1}$, which might have been influenced by the strong signal from bound/coordinated water OH ($v_s$ and $v_{as}$ stretching) is high in magnitude (approx. 30.0 kJ mol$^{-1}$ and approx. 40.1 kJ mol$^{-1}$). At high wavenumber, 3554 cm$^{-1}$, there is a decrease in H-bond energy with growth duration (downwards in the column in table 3), from 7.2 kJ mol$^{-1}$ to 6.6 kJ mol$^{-1}$, for *T. versicolor*, indicating lesser accessibility of free water and scissoring of stronger interactions. At lower wavenumbers, i.e. 3229 cm$^{-1}$ and 3084 cm$^{-1}$, there are no significant changes for *T. versicolor* indicating stronger presence of bound water. The FTIR spectra for *P. placenta* showed relatively no change in bond energy through the growth period (table 3).

A paired- '$t$' test was performed assuming equal mean for the pair of fungal species to evaluate the statistical difference between growth periods (2, 4, 6 and 8 weeks) versus absorbance wavenumbers (3554 cm$^{-1}$, 3390 cm$^{-1}$, 3229 cm$^{-1}$ and 3084 cm$^{-1}$). From table 3, the calculated column-wise comparison gives $p$-values as 0.354, 0.043, 0.763 and 0.190, respectively for 3554 cm$^{-1}$, 3390 cm$^{-1}$, 3229 cm$^{-1}$ and 3084 cm$^{-1}$ wavenumbers. Thus, it can be statistically inferred that H-bond energy at 3390 cm$^{-1}$, linked with free water ($v_s$ stretching [43]) differ significantly ($p = 0.043$), in IR bands obtained from *P. placenta* being comparatively higher in magnitude, at four observation periods. There is no statistically significant difference observed for different growth periods in H-bond energies at other wavenumbers. Moreover, from table 3, the calculated row-wise comparison yields $p$-values as 0.108, 0.861, 0.474 and 0.900. Therefore, it may be concluded that there is no statistically significant difference between H-bond energies observed individually in the growth period at 2nd, 4th, 6th and 8th weeks for the studied pair of fungi.

## 3.5. Hydrogen bond distance

Table 4 shows that the H-bond distance is nearly unchanged during the period of evaluation. The calculated average H-bond distances at the four different zones of deconvoluted IR bands were 2.83 Å, 2.79 Å, 2.76 Å and 2.72 Å, respectively. Therefore, it can be said that with decreasing wavenumber the hydrogen bond distance decreases. Furthermore, as seen in §3.3., the H-bond distance at higher wavenumbers can be associated with free water and lower wavenumbers are associated with bound water. Therefore, table 4 shows that the H-bond distance of free water ($v_{as}$ stretching) is longer in dimension than the H-bond linked with bound water ($v_s$ and $v_{as}$ stretching) in the cell. Moreover, comparing tables 3 and 4, shows that as the bond distances (Å) decreases, the H-bond energies ($E_H$,

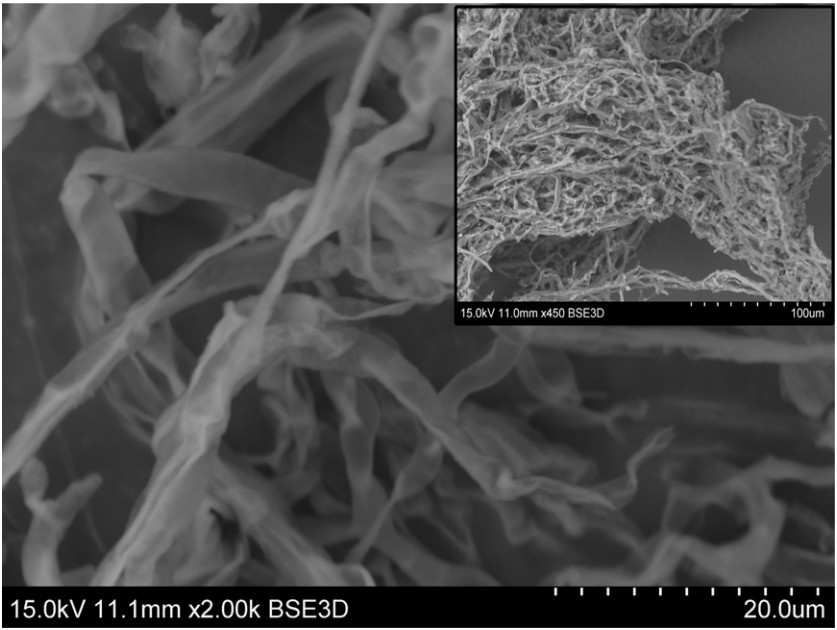

**Figure 8.** Scanning electron microscopic image of hyphae of the brown rot fungus, *Postia placenta*.

**Table 4.** Hydrogen bond distances (Å) at different wavenumbers (cm$^{-1}$) at shifting frequencies at various growth times.

| species | growth (weeks) | hydrogen bond distance (Å)[a] | | | |
| --- | --- | --- | --- | --- | --- |
| | | 3554 cm$^{-1}$ | 3390 cm$^{-1}$ | 3229 cm$^{-1}$ | 3084 cm$^{-1}$ |
| *Trametes versicolor* | 2 | 2.8285 ± 0.0003 | 2.792 ± 0.001 | 2.757 ± 0.001 | 2.7233 ± 0.0007 |
| | 4 | 2.8296 ± 0.0008 | 2.7929 ± 0.0007 | 2.756 ± 0.001 | 2.7234 ± 0.0005 |
| | 6 | 2.830 ± 0.002 | 2.7931 ± 0.0007 | 2.757 ± 0.003 | 2.7239 ± 0.0007 |
| | 8 | 2.831 ± 0.002 | 2.7928 ± 0.0004 | 2.7554 ± 0.0007 | 2.7213 ± 0.0004 |
| *Postia placenta* | 2 | 2.829 ± 0.001 | 2.7904 ± 0.0008 | 2.755 ± 0.002 | 2.7235 ± 0.0008 |
| | 4 | 2.8285 ± 0.0003 | 2.791 ± 0.001 | 2.757 ± 0.002 | 2.7243 ± 0.0001 |
| | 6 | 2.8298 ± 0.0009 | 2.7928 ± 0.0009 | 2.7559 ± 0.0004 | 2.724 ± 0.001 |
| | 8 | 2.8285 ± 0.0002 | 2.792 ± 0.002 | 2.757 ± 0.001 | 2.7248 ± 0.0004 |

[a]Mean of four measurements ± 2 s.d. of the mean (95.45% confidence interval).

kJ mol$^{-1}$) increases. Other researchers observed that, in a H-bond environment of several protein amino acids and a water molecule, the hydroxyl oxygen is 2.98 Å away from a carbonyl (C=O) oxygen and 2.7 Å from the oxygen atom of the water (H$_2$O) molecule [33]. The repulsion between lone pair electrons, residing on the oxygen atoms, could be the reason for low H-bond energy (average 6.8–7.2 kJ mol$^{-1}$) and stretching of H-bonds (average 2.83 Å) observed at deconvoluted IR band 3554 cm$^{-1}$ compared to results obtained from other deconvoluted bands. The reported H-bond distance in wood cellulose varies from 2.83 Å at IR band 3567 cm$^{-1}$ to 2.76 Å at IR band 3252 cm$^{-1}$ [45]. In wood, the H-bond energy and the H-bond distances at 3221 cm$^{-1}$ were 30.35 kJ mol$^{-1}$ and 2.75 Å, respectively, as observed by other researchers [46]. In lignin, the H-bond distances have been found to vary from 1.78 Å to 2.34 Å [47]. Therefore, it can be concluded that the H-bond distances calculated from the deconvolution of the H-N and O-H group frequencies are within the acceptable range of previous studies.

## 3.6. Morphology

Morphology of the mycelial growth was characterized by SEM images. Figure 8 shows the hyphal structure of *P. placenta*. It has been revealed by previous researchers that hyphal tip extension is

dominant during the initial stages of the growth process [48]. Increase in mycelium density during the later stages was accompanied by lateral hyphal branching. As hyphal branching increased along with tip extension, the mycelium spread from colonized to uncolonized regions rapidly. Tip extension is believed to occur via inner imbibed pressure and structural wall fibril synthesis, as have been found by previous researchers. As discussed in §3.2, there is an increase in cellular protein content, amide I band at 1636 cm$^{-1}$ and amide II band at 1546 cm$^{-1}$, observed during the 2–8 weeks growth period. Furthermore, the weak H-bond energy at 3390 cm$^{-1}$, linked with free water ($v_s$ stretching), is high from the cell structure of *P. placenta* which might be linked with imbibed wall pressure and hyphal wall fibril synthesis.

# 4. Conclusion

In this study, cell structures of the two wood decay fungi, namely, *T. versicolor* and *P. placenta*, grown on Norway spruce wood were evaluated by ATR-FTIR spectroscopy at 2nd, 4th, 6th and 8th weeks of growth with the following observations:

(i) the statistical analysis shows that there is no significant relationship in decay/mass loss in wood substrate caused by the two species of fungi;

(ii) common IR bands: amide I band at 1636 cm$^{-1}$ and amide II band at 1546 cm$^{-1}$; cell lipids show band at 1465 cm$^{-1}$ for CH$_2$ bending mode; band at 1402 cm$^{-1}$ for C=O symmetric stretching of COO$^-$ groups; band at 1377 cm$^{-1}$ for C-H bending mode of CH$_2$; 1030 cm$^{-1}$ is assigned to the C-O deformation in primary alcohol; phospholipids and PO$_2$ show bands at 1237 cm$^{-1}$ and 1082 cm$^{-1}$;

(iii) IR bands assigned to *T. versicolor*: protein α-helices at 1655 cm$^{-1}$; protein random coils at 1644 cm$^{-1}$; protein ß-sheet at 1636 cm$^{-1}$; histidine amino acid, 4-methylimidazole, at 1633 cm$^{-1}$; amino acid arginine symmetric stretching vibration near 1635 cm$^{-1}$; α-helices of amide III at 1283 cm$^{-1}$; carbohydrate C-C bond at 1036 cm$^{-1}$; primary alcoholic $v$(C−O) at 1020 cm$^{-1}$;

(iv) IR bands assigned to *P. placenta*: protein α-helices at 1654 cm$^{-1}$; protein random coil at 1644 cm$^{-1}$; ß sheet at 1625 cm$^{-1}$; amino acid arginine antisymmetric band region 1695–1652 cm$^{-1}$; amino acid arginine symmetric band region of 1663–1614 cm$^{-1}$; protein amino acid lysine asymmetric band region of 1629–1626 cm$^{-1}$; protein α-helices and ß-sheet structure at 1558 cm$^{-1}$ - 1542 cm$^{-1}$; protein glutamic acid antisymmetric band in the region 1560–1556 cm$^{-1}$; amide III at 1310 cm$^{-1}$; primary alcohol C−OH group at 1030 cm$^{-1}$; two shoulders of acid carboxyl group at 1700 cm$^{-1}$ and 1686 cm$^{-1}$; amino acid tryptophan stretching vibrations at 1505 cm−1;

(v) a steady increase in protein and carbohydrate content in the fungal cell material during the growth period; and

(vi) deconvolution of 3600–3050 cm$^{-1}$ broad OH region shows presence of four bands at 3554 cm$^{-1}$, 3390 cm$^{-1}$, 3229 cm$^{-1}$ and 3084 cm$^{-1}$ which provides further estimations for the H-bond energy and the H-bond distance. The H-bond energies at the four bands were in the range of 6.8–40.5 kJ mol$^{-1}$. The H-bond energy at higher wavenumbers is less in magnitude than the bond energy at the lower wavenumbers, indicating stronger interactions at lower wavenumber. The H-bond distances at the four different deconvoluted bands were found to be in decreasing order of magnitude 2.83 Å, 2.79 Å, 2.76 Å and 2.72 Å, respectively, at decreasing wavenumbers. The sequence of the H-bond distance reaffirmed the findings that at lower wavenumber, the interactions are stronger. The microscopic study further confirmed the findings of FTIR spectra by showing the presence of hyphal tip and mycelium.

The consistency of spectral findings that were obtained from the (i) raw data, (ii) 2nd derivative, and the (iii) Fourier deconvolution, shows the efficiency of ATR-FTIR spectroscopy in analysing microbial cell *in vivo* conditions that confirmed the findings of microscopic images. Further studies may be done to integrate the spectral data, from microbial cells, with the time-related microscopic studies.

Data accessibility. Our data are deposited in Dryad Digital Repository: https://doi.org/10.5061/dryad.6hdr7sqxf [49].
Authors' contributions. B.S.G.: conceptualization, data curation, formal analysis, investigation, methodology, validation, writing—original draft, writing—review and editing; B.P.J.: conceptualization, funding acquisition, methodology, project administration, resources, software, supervision, writing—review and editing; T.G.: validation. All authors gave final approval for publication and agreed to be held accountable for the work performed therein.
Competing interests. We declare we have no competing interests.

Funding. The authors gratefully acknowledge the financial support from the Research Council of Norway; NTNU Trondheim; SINTEF Trondheim; Viken Skog BA; Treindustrien; the Wood Technology Research Fund at the Norwegian Institute of Wood Technology; Jotun AS and Kebony ASA.

Acknowledgements. This research work was performed at the laboratory of NTNU and SINTEF Trondheim, Norway, under the project titled 'Service life estimation of wood façade'. The authors are grateful to Prof Emeritus Dr Per Jostein Hovde from NTNU, Trondheim, for valuable suggestions; Dr Gry Alfredsen and Dr Lone Gobakken, from the Norwegian Forest and Landscape Institute at Ås, for the fungi strains. Further, the authors would like to thank the research partners of the project, i.e. the Norwegian University of Life Sciences (UMB), the Norwegian Forest and Landscape Institute (Skog og Landskap) and the Norwegian Institute of Wood Technology (Treteknisk).

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
