## [Peer Review File · Royal Society Open Science]

Review History

RSOS-200291.R0 (Original submission)

Review form: Reviewer 1

Is the manuscript scientifically sound in its present form?

No

Are the interpretations and conclusions justified by the results?

No

Is the language acceptable?

Yes

Do you have any ethical concerns with this paper?

No

Have you any concerns about statistical analyses in this paper?

No

Recommendation?

Accept with minor revision (please list in comments)

Comments to the Author(s)

- o Can the authors link changes in the amino acids and internal structure of the two fungi to any potential physiology changes in the two fungi?
- o Was the SEM used required drying of the fungal specimens? If so, would influence your results?

Review form: Reviewer 2**Is the manuscript scientifically sound in its present form?**

No

Are the interpretations and conclusions justified by the results?

Yes

Is the language acceptable?

Yes

Do you have any ethical concerns with this paper?

No

Have you any concerns about statistical analyses in this paper?

No

Recommendation?

Reject

Comments to the Author(s)

This study describes monitoring of fungal growth during wood degradation by using FTIR spectroscopy.

Authors collected the spectra at 2 weeks interval to understand the behavior of fungi and discuss hydrogen bond energy.

Though the experiments have been well performed and discussion is an interesting, present version can not be accepted because there is little information on substrates such as weight loss rate and chemical component analysis.

SEM observation is unclear to understand the hyphal tip extension from single snapshot.

Despite critical comments, IR spectral monitoring of fungi during bio-degradation involves valuable information for readers. Therefore, reviewer hopes the document will be modified and improved for publish.

Decision letter (RSOS-200291.R0)

Dear Dr Gupta:

Manuscript ID RSOS-200291 entitled "Evaluating the scope of infrared spectroscopy in periodic cellular characterization of wood decay fungi" which you submitted to Royal Society Open Science, has been reviewed. The comments from reviewers are included at the bottom of this letter.

In view of the criticisms of the reviewers, the manuscript has been rejected in its current form. However, a new manuscript may be submitted which takes into consideration these comments.

Please note that resubmitting your manuscript does not guarantee eventual acceptance, and that your resubmission will be subject to peer review before a decision is made.

Your resubmitted manuscript should be submitted by 22-Dec-2020. If you are unable to submit by this date please contact the Editorial Office.

on behalf of Prof R. Kerry Rowe (Subject Editor)
openscience@royalsociety.org

Associate Editor Comments to Author:

The editors would like you to resubmit a revised version of your paper to take into account the critiques of the reviewers. Please be aware that the editors may seek additional refereeing advice upon resubmission.

Reviewers' Comments to Author:

Reviewer: 1

Comments to the Author(s)

o Can the authors link changes in the amino acids and internal structure of the two fungi to any potential physiology changes in the two fungi?
o Was the SEM used required drying of the fungal specimens? If so, would influence your results?

Reviewer: 2

Comments to the Author(s)

This study describes monitoring of fungal growth during wood degradation by using FTIR spectroscopy.

Authors collected the spectra at 2 weeks interval to understand the behavior of fungi and discuss hydrogen bond energy.

Though the experiments have been well performed and discussion is an interesting, present version can not be accepted

because there is little information on substrates such as weight loss rate and chemical component analysis.

SEM observation is unclear to understand the hyphal tip extension from single snapshot.

Despite critical comments, IR spectral monitoring of fungi during bio-degradation involves valuable information for readers. Therefore, reviewer hopes the document will be modified and improved for publish.

Author's Response to Decision Letter for (RSOS-200291.R0)

See Appendix A.

RSOS-201935.R0

Review form: Reviewer 2

Is the manuscript scientifically sound in its present form?

Yes

Are the interpretations and conclusions justified by the results?

Yes

Is the language acceptable?

Yes

Do you have any ethical concerns with this paper?

No

Have you any concerns about statistical analyses in this paper?

No

Recommendation?

Accept as is

Comments to the Author(s)

The revised manuscript will be accepted.

Review form: Reviewer 3

Is the manuscript scientifically sound in its present form?

Yes

Are the interpretations and conclusions justified by the results?

Yes

Is the language acceptable?

Yes

Do you have any ethical concerns with this paper?

No

Have you any concerns about statistical analyses in this paper?

No

Recommendation?

Accept with minor revision (please list in comments)

Comments to the Author(s)

Dear Editor, Dear Authors

Please find my observation regarding the manuscript entitled: "In vitro cell composition identification of wood decay fungi by Fourier transform infrared spectroscopy" you requested to revise. The manuscript presents the application of ATR FTIR for analysing of molecular compositions the fungi cell structure. The manuscript is clearly written and easy to follow, I just have a few comments/observations that authors might consider for its improvement.

Why was the experiment conducted for 8 weeks? Does it refer to any standard? Please explain.

In line 15-31 you write: "In natural outdoor conditions, the process of fungal degradation of wood is a slow process and often do not show any visible discolouration, at the onset, causing difficulty in judgement of the soundness of wood. Contrarily, in accelerated laboratory conditions, using - (a) sterile materials, (b) nutrients, (c) monoculture, and (d) moisture, fungi grow faster than in natural environment. A study of fungal cell structure in accelerated laboratory conditions, therefore, provides meaningful information about the molecular compositions in microbial cell structure (5)."

Is it only the speed of growth different, or also kinetic? Please explain how the laboratory experiment can help the understanding of the processes occurring in nature.

The analysis of spectra is correct however rather a standard one. Much more information can be revealed by some chemometrics methods. I highly recommend using 2D spectral correlation analysis that enables the study of molecular-level changes induced by an external perturbation (time in this case). Perhaps you might consider this approach in your future analysis. Please correct some punctuation errors.

Decision letter (RSOS-201935.R0)

Dear Dr Gupta

On behalf of the Editors, we are pleased to inform you that your Manuscript RSOS-201935 "In vitro cell composition identification of wood decay fungi by Fourier transform infrared spectroscopy" has been accepted for publication in Royal Society Open Science subject to minor

revision in accordance with the referees' reports. Please find the referees' comments along with any feedback from the Editors below my signature.

Please submit your revised manuscript and required files (see below) no later than 7 days from today's (ie 22-Apr-2021) date. Note: the ScholarOne system will 'lock' if submission of the revision is attempted 7 or more days after the deadline. If you do not think you will be able to meet this deadline please contact the editorial office immediately.

on behalf of Professor R. Kerry Rowe (Subject Editor)
openscience@royalsociety.org

Associate Editor Comments to Author:

Thank you for engaging with the reviewers' concerns. While the paper is much nearer publication-readiness than the initial iteration, there are a number of outstanding queries that you should address before resubmitting a final version.

Reviewer comments to Author:

Reviewer: 2
Comments to the Author(s)
The revised manuscript will be accepted.

Reviewer: 3
Comments to the Author(s)
Dear Editor, Dear Authors

Please find my observation regarding the manuscript entitled: "In vitro cell composition identification of wood decay fungi by Fourier transform infrared spectroscopy" you requested to revise. The manuscript presents the application of ATR FTIR for analysing of molecular compositions the fungi cell structure. The manuscript is clearly written and easy to follow, I just have a few comments/observations that authors might consider for its improvement.

Why was the experiment conducted for 8 weeks? Does it refer to any standard? Please explain. In line 15-31 you write: "In natural outdoor conditions, the process of fungal degradation of wood is a slow process and often do not show any visible discolouration, at the onset, causing difficulty in judgement of the soundness of wood. Contrarily, in accelerated laboratory conditions, using - (a) sterile materials, (b) nutrients, (c) monoculture, and (d) moisture, fungi grow faster than in natural environment. A study of fungal cell structure in accelerated laboratory conditions, therefore, provides meaningful information about the molecular compositions in microbial cell structure (5)."

Is it only the speed of growth different, or also kinetic? Please explain how the laboratory experiment can help the understanding of the processes occurring in nature.

The analysis of spectra is correct however rather a standard one. Much more information can be revealed by some chemometrics methods. I highly recommend using 2D spectral correlation analysis that enables the study of molecular-level changes induced by an external perturbation (time in this case). Perhaps you might consider this approach in your future analysis.

Please correct some punctuation errors.

===PREPARING YOUR MANUSCRIPT===

===PREPARING YOUR REVISION IN SCHOLARONE===

Author's Response to Decision Letter for (RSOS-201935.R0)

See Appendix B.

Decision letter (RSOS-201935.R1)

Dear Dr Gupta,

I am pleased to inform you that your manuscript entitled "In vitro cell composition identification of wood decay fungi by Fourier transform infrared spectroscopy" is now accepted for publication in Royal Society Open Science.

Kind regards,
Royal Society Open Science Editorial Office
Royal Society Open Science

on behalf of R. Kerry Rowe (Subject Editor)
openscience@royalsociety.org

Appendix A

To
Dr. Andrew Dunn
Royal Society Open Science Editorial Office
Royal Society Open Science
openscience@royalsociety.org

27th Oct 2020

Subject: Manuscript RSOS-200291

Hello Dr. Dunn

Refer to your journal's decision, communicated to us on 24 June 2020, we have done the manuscript revision as per the comments received from the reviewers:

Reviewer 1

Can the authors link changes in the amino acids and internal structure of the two fungi to any potential physiology changes in the two fungi?

The Abstract, Discussion and Conclusion section has been revised and modified to incorporate the details of physiological changes observed in the two fungi.

Was the SEM used required drying of the fungal specimens? If so, would influence your results?

In the methodology section, we have mentioned that we collected backscatter image avoiding charge build-up in the living cell specimens.

Reviewer 2

there is little information on substrates such as weight loss rate and chemical component analysis.

The discussion section has been rephrased and rewritten to incorporate informations regarding chemical component changes. Additionally, the Conclusion section has been rewritten to provide more information

SEM observation is unclear to understand the hyphal tip extension from single snapshot.

We cited a reference that says that propagation occurs via tip extension. Our future plan was to perform spectral mapping correlating with microscopic observations.

Despite critical comments, IR spectral monitoring of fungi during bio-degradation involves valuable information for readers. Therefore, reviewer hopes the document will be modified and improved for publish.

Thank you. We appreciate your critical comments.

Looking forward to get the article published in your journal soon

On behalf of all authors,

Kind regards

Barun Shankar Gupta (PhD,M.S.,MSc)

Editorial board member of *International Journal of Materials Science and Applications*

Norwegian University of Science and Technology (NTNU),

Høgskoleringen 7A, NO-7491 Trondheim, Norway.

Phone: +47 73594640, +91-8902790048

Fax : +47 73 59 70 21

bsbarun@gmail.com

https://www.researchgate.net/profile/Barun_Shankar_Gupta

Appendix B

Reviewer 2

The revised manuscript will be accepted.
The manuscript has been revised as per comments.

Reviewer 3

Why was the experiment conducted for 8 weeks? Does it refer to any standard? Please explain.

We have rephrased the paragraph. The new paragraph reads “*From the above discussions, a hypothesis can be drawn that the cellular structure of wood fungi depends on the duration of wood deterioration process and 70-84 days of microbial exposure is sufficient for the wood cell wall material to get degraded. Henceforth, the aim of this study is to perform inter-species investigation of cell structure during the 8-week period of wood decay process through monoculture*”.

Is it only the speed of growth different, or also kinetic? Please explain how the laboratory experiment can help the understanding of the processes occurring in nature.?

We have rephrased the sentence. The new sentence reads “*Contrarily, in accelerated laboratory conditions, using – (a) sterile materials, (b) nutrients, (c) monoculture, and (d) moisture, the interspecies antagonistic invasion is prevented thereby allowing the microbial cells to follow normal metabolic and enzyme activities.*”

The analysis of spectra is correct however rather a standard one. Much more information can be revealed by some chemometrics methods. I highly recommend using 2D spectral correlation analysis that enables the study of molecular-level changes induced by an external perturbation (time in this case). Perhaps you might consider this approach in your future analysis.

Noted.

Please correct some punctuation errors.

We have run the Spelling & Grammar error check options, and wherever recommended have accepted the grammatical mistake corrections. Thank you. We appreciate your critical comments.